# Elevated Levels of Monocyte Chemotactic Protein-1 in the Follicular Fluid Reveals Different Populations among Women with Severe Endometriosis

**DOI:** 10.3390/jcm9051306

**Published:** 2020-05-01

**Authors:** Pierre-Emmanuel Bouet, Juan-Manuel Chao de la Barca, Lisa Boucret, Philippe Descamps, Guillaume Legendre, Hady El Hachem, Simon Blanchard, Pascale Jeannin, Pascal Reynier, Pascale May-Panloup

**Affiliations:** 1Département de Médecine de la Reproduction, Centre Hospitalier Universitaire, 49000 Angers, France; phdescamps@chu-angers.fr (P.D.); guillaume.legendre@chu-angers.fr (G.L.); 2Unité Mixte de Recherche MITOVASC, Equipe Mitolab, Centre National de la Recherche Scientifique 6015, Institut National de la Santé et de la Recherche Médicale U1083, Université d’Angers, 49000 Angers, France; juanmanuel.chaodelabarca@univ-angers.fr (J.-M.C.d.l.B.); liboucret@chu-angers.fr (L.B.); pareynier@chu-angers.fr (P.R.); pamaypanloup@chu-angers.fr (P.M.-P.); 3Département de Biochimie et Génétique, Centre Hospitalier Universitaire, 49000 Angers, France; 4Département de Biologie de la Reproduction, Centre Hospitalier Universitaire, 49000 Angers, France; 5Département de Médecine de la Reproduction, Centre Médical Clemenceau, 1103 Beyrouth, Liban; hadyhachem@hotmail.com; 6Département d’Immunologie et d’Allergologie, Centre Hospitalier Universitaire, 49000 Angers, France; simon.blanchard@univ-angers.fr (S.B.); pascale.jeannin@univ-angers.fr (P.J.); 7Unité Mixte de Recherche CRCINA, Institut National de la Santé et de la Recherche Médicale, Université d’Angers, 49000 Angers, France

**Keywords:** follicular fluid, cytokines, endometriosis, oocyte, MCP-1

## Abstract

To determine if a modification of the cytokine profile occurs in the follicular fluid (FF) of women with endometriosis undergoing in vitro fertilization (IVF), we performed a prospective observational study from January 2018 to February 2019. In total, 87 women undergoing IVF were included: 43 for severe endometriosis-related infertility and 40 controls with other causes of infertility. The cytokine profile of the FF was determined by multiplex fluorescent-bead-based technology allowing the measurement of 59 cytokines. Monocyte Chemoattractant Protein 1 (MCP-1) was the only variable retained in the multivariate analysis. We identified two subgroups of patients in the endometriosis group: MCP-1-low group (*n* = 23), which had FF MCP-1 levels comparable to the control group, and MCP-1-high (*n* = 20), which had significantly higher FF levels. Only patients in the MCP-1-high group had a significantly altered cytokine profile in the FF, and had a significantly higher serum estradiol level (*p* = 0.002) and a significantly lower number of oocytes recovered (*p* = 0.01) compared to the MCP-1-low and the control group. Our study has shown an alteration of the oocyte microenvironment in women with endometriosis associated with high follicular fluid levels of MCP-1, allowing the identification of a subgroup of endometriosis patients with a potentially worse prognosis.

## 1. Introduction

Endometriosis is a chronic disease affecting 10% of women of reproductive age [1]. Women with endometriosis often present with dysmenorrhea and pelvic pain, dyspareunia, and in 25% of cases, infertility [2,3]. The mechanisms by which endometriosis causes infertility are not yet clearly established, but several factors have been incriminated [1]. Endometriosis could have a negative impact on folliculogenesis, ovulation, oocyte quality, tubal mobility, and implantation, with several mechanical, immunological, genetic, and environmental factors implicated [4].

The direct impact of endometriosis on oocyte quality has been demonstrated in studies using the oocyte donation model. Hauzman et al. showed that implantation rates were significantly lower in recipients without evidence of endometriosis who received oocyte donation from women with endometriosis compared to recipients with endometriosis who received oocytes from women without endometriosis [5]. However, four meta-analyses on the subject gave conflicting results [6,7,8,9], and data are still insufficient to confirm the direct link and the mechanisms behind it [10]. Indeed, various elements could affect these results, such as the stage of the disease, the previous medical or surgical treatment history, and the presence of other comorbidities that could impact fertility, independently of endometriosis.

Several factors implicated in the acute and chronic inflammatory reaction (hormones, cytokines, chemokines, and markers of oxidative stress) are thought to be involved in the pathophysiology of endometriosis. Indeed, an alteration of the cytokine profile has been shown in the serum, the peritoneal fluid, the endometrial tissue, and in endometriomas of women with endometriosis [11,12]. Recent studies have focused on the follicular fluid (FF) composition as a means of assessing the oocyte’s—and the resulting embryo’s—quality [13], since it reflects the exchanges occurring between the oocyte and its microenvironment during the acquisition of gametic competence along oogenesis and folliculogenesis [14]. Different biological signatures have been assessed in order to better understand the pathophysiology behind the endometriosis-related infertility, using proteomics [15], metabolomics [16,17,18,19,20], and the analysis of the cytokine profile [21,22,23,24,25]. However, such studies have been scarce and have yielded disparate results.

Therefore, the aim of our study was to compare the cytokine profiles of the FF in patients undergoing in vitro fertilization (IVF) for isolated forms of severe endometriosis to those of patients free from endometriosis, and to evaluate the potential impact of the disease on the oocyte microenvironment.

## 2. Materials and Methods

### 2.1. Study Participants 

Participants were included in the study from January 2018 to February 2019 after having given their informed written consent for the research. The study was conducted according to the ethical standards of the Helsinki Declaration and its later amendments, and with the approval of the Ethics Committee of the University Hospital of Angers, France (Number DC-2014-2224 and AC-2016-2799).

We included only patients with severe endometriosis according to the classification of the American Society of Reproductive Medicine [2], i.e., having an endometrioma and classified as stage III or stage IV. The diagnosis of endometriosis had been previously confirmed by ultrasound or Magnetic Resonance Imaging (MRI), or following abdomino-pelvic surgery. Women without endometriosis, undergoing IVF for other indications (unexplained, male factor, tubal factor), were included as the control group. Patients in both groups had either IVF or intracytoplasmic sperm injection (ICSI), depending on the indication. At our center, we perform IVF in cases of tubal factor or unexplained infertility, and ICSI in cases with a previous history of failed or suboptimal fertilization (fertilization rate < 20%), and in cases with severe male factor infertility. All patients aged between 18 and 43 years could be included in the study.

We excluded women with polycystic ovarian syndrome, a history of cancer, and premature ovarian failure, since these pathologies could impact the FF microenvironment.

All patients had controlled ovarian stimulation (COS), using either the long agonist or the antagonist protocol. At our center, we offer the long agonist protocol in the first line for patients with endometriosis, based on studies showing better pregnancy rates [26,27]. Both recombinant and urinary gonadotropins were used, based on the treating physician’s discretion. All patients had serial ultrasound and blood monitoring, and when ≥3 follicles ≥17 mm were noted, ovulation was triggered with 250 μg recombinant HCG (Ovitrelle^®^, Merck, Lyon, France), and oocyte retrieval was performed under ultrasound guidance 36 h later. Patients were enrolled during the morning of oocyte retrieval.

### 2.2. Embryo Culture and FF Samples

After IVF, embryos were cultivated in Global medium^®^ (Life Global, Guilford, CT, USA) under an atmosphere of CO_2_. Embryos were observed 18 h post-fertilization to objectify the presence of the two pronuclei, and 48 h post-fertilization to objectify embryo cleavage. The embryos were scored at day 2 (48 h post-injection) according to the European Society of Human Reproduction and Embryology (ESHRE) consensus [28] defining three groups of embryos, i.e., “good quality,” “fair quality,” or “poor quality” embryos. 

Once the oocytes were isolated for fertilization and culture, FF samples were collected, pooled, and immediately centrifuged for 10 min at 3000 g at +4 °C. The supernatant was recovered and conserved at −80 °C in 500 μL aliquots until the analysis.

### 2.3. Quantification of Cytokines and other Inflammation-Aassociated Ffactors

The cytokinome was determined in the FF by multiplex fluorescent-bead-based technology (Luminex 200™, Austin, TX, USA), using two commercial Luminex screening assay kits (Bio-Plex Pro™ Human Cytokine 27-plex Assay and Bio-Plex Pro Human Inflammation Panel 1, 37-plex Bio-Rad) from Bio-Rad Laboratories (Marnes-la-Coquette, IDF, France), allowing the measurement of 57 cytokines and other inflammation-associated factors (Matrix MetalloProteinase-3 (MMP-3), Chitinase-3 like 1 protein). Briefly, the FF samples were diluted two-fold before incubation with specific antibody-coated fluorescent beads according to the manufacturer’s recommendations. After washing, 50 beads were analyzed with the Luminex 200™ analyzer and Bio-Plex Manager software version 6 (Bio-Rad Laboratories, Hercules, CA, USA, 2010), and the cytokine concentrations of the samples were estimated through the serial dilution of cytokine standards.

### 2.4. Statistical Analysis 

Cytokine concentrations were log-transformed before the application of Student’s test. Benjamini-Hochberg correction was applied to correct for type-I inflation due to test multiplicity, and to keep false discovery rate under 5%. All data were divided in two sets: training-validation (~ 2/3 of all data) and test (~1/3 of all data) sets. Least Absolute Shrinkage and Selection Operator (LASSO) and logistic regression (LR) were used for model and variable selection in the training-validation set. To avoid selecting overfitted models, the predictive performance of the selected model was evaluated for predicting group allocation (i.e., endometriosis or control) of each sample of the test set. The Area Under the Receiver Operating Characteristic Curve (AUROC) and permutation test were used as a metric evaluating predictive capabilities of the multivariable models, with models having AUROC > 0.8 considered as models with good predictive capabilities. This multivariate model integrates cytokines concentrations along with clinical variables acting as potential confounders (i.e., varying significantly between control and women suffering from endometriosis). More detailed information about the multivariate analysis is provided in Figure 1.

## 3. Results

In total, 87 women undergoing IVF were included, 43 with severe endometriosis (endometriosis group) and 44 without endometriosis (control group). The patients’ characteristics (age, body mass index (BMI), tobacco use, hormonal profile, antral follicle count) were comparable between the endometriosis and control groups (Table 1). 

The baseline characteristics of cycles are described in Table 1. As expected, significantly more patients received the long protocol in the endometriosis group compared to the control group (*p* < 0.001). Moreover, in the endometriosis group, the total Follicule Stimulating Hormone (FSH) dose received was significantly higher (2810 ± 1072 vs. 2375 ± 900 International Unit (IU), *p* = 0.04), and the total number of oocytes retrieved was significantly lower (8.8 ± 7.0 vs. 13.4 ± 5.9, *p* = 0.002) than the control group. However, there was no difference in the fertilization rate and the rate of good embryos. 

### 3.1. Cytokine Analysis

Overall, the Granulocyte Macrophage-Colony Stimulating Factor (GM-CSF) and IL-15 levels obtained were uninterpretable (zero), thus, we ended up including 57 compounds in our final analysis. 

Univariate analysis: after Benjamini-Hochberg correction, the concentrations of nine cytokines (Monocyte Chemoattractant Protein 1 (MCP-1), also referred to as chemokine ligand 2 (CCL2), Interleukin-6 (IL-6), IL-8, IL-1b, IL-5, chitinase 3-like 1 protein, osteocalcin, MMP-3 and basic Fibroblast Growth Factor (FGF), also known as FGF2) were found to be significantly higher in the FF of patients with endometriosis compared to controls (Figure 2).

Multivariate analysis: After applying the LASSO to the training-validation set, only two variables (“Downregulation protocol” and MCP-1) had non-null coefficients. Backward LR confirmed the importance of these two variables as the model with only one of these variables had significantly less predictive capabilities compared to the complete model. Interaction between these variables was not retained, as it only increased the model complexity without increasing the predictive capabilities. Histograms of AUROCs for the training and the validation set are represented in the Figure 3A. 

The final LR model after logit transformation was: LogitP (Endometriosis Downregulation protocol, sMCP-1) = −2.50 + 2.53*(Downregulation protocol) + 2.04*(MCP-1)

It should be noted that MCP-1, as well as all other quantitative variables, were centered (null mean) and scaled to unit variance before the LASSO and LR. 

When this model was applied to the test set, the AUROC obtained was 0.95 (Figure 3B). Permuting the response vector in the test set and applying the precedent LR model yielded predictions identical to those of the random model with null predictive capabilities (Figure 3C).

### 3.2. Identification of two Subgroups of Patients

While analyzing the MCP-1 levels in the FF of women with endometriosis, we noticed that some patients had levels equivalent to those of the control group, while others had significantly higher levels. Therefore, we subdivided the women in the endometriosis group into two subgroups based on the MCP-1 levels in the FF, using the maximal level found in the control group as the cutoff point: 350 pg/mL. We had two subgroups: the MCP-1-low group (*n* = 23) had levels < 350 pg/mL, and the MCP-1-high group (*n* = 20) had levels > 350 pg/mL (Figure 4). After Benjamini-Hochberg correction, the concentrations of 11 cytokines (the same ones found to be different between the two groups barring osteocalcin, as well as Granulocyte-Colony Stimulating Factor (GCSF), soluble Tumor Necrosis Factor Receptor type 1 (sTNF_R1), and B-cell Activating Factor (BAFF)) were found significantly higher in the FF of MCP-1-high patients compared to MCP-1-low patients. The analysis of the inflammatory profile of these two subgroups, when compared to the control group, showed a dysregulation only in the MCP-1-high subgroup.

Moreover, the subgroup analysis showed a significantly lower number of oocytes retrieved (6 ± 3.6 vs. 11.3 ± 8.3, *p* = 0.01) and a significantly higher basal estradiol level (82.1 ± 67.2 vs. 32.6 ± 12.9, *p* = 0.002) in the MCP-1-high group when compared to the MCP-1-low group. The results were the same when comparing the MCP-1-high group to the control group (*p* < 0.001 for the two variables), but there were no differences between the MCP-1-low and the control group. The number of top-quality embryos was lower in the MCP-1-high group (49%) compared to MCP-1-low group (65%), but the difference was not statistically different (*p* = 0.1). 

Finally, concerning the characteristics of endometriosis (locations, adhesions, diagnosis method) and the history of surgery, there were no significant differences between the MCP-1-high and the MCP1-low groups (Table 2).

## 4. Discussion

### 4.1. General Considerations

The current study has shown an alteration of the cytokine profile in the follicular fluid of women with endometriosis, with high levels of MCP-1 linked to an alteration of the oocyte microenvironment.

Several studies have assessed the inflammatory profile of the peritoneal fluid, the ectopic implantation tissues, and the serum in women with endometriosis, and most have found a global cytokine activation, with an overexpression of both pro and anti-inflammatory cytokines [11,12]. The most specific activated cytokines were IL6, TNFα, IL8, IL1β, MCP-1, and Regulated on Activation Normal T cell Expressed and Secreted (RANTES) [11]. Based on these findings, we decided to analyze the inflammatory profile of the FF, since it is a direct reflection of the oocyte micro-environment, and plays a crucial role in the acquisition of gamete competence. However, to date, cytokine studies of FF have given conflicting results, probably due to the heterogeneity in the populations included and the cytokines analyzed (mostly less than 10), and the presence of several confounding factors [21,22,23,24,25]. Indeed, many factors and comorbidities can impact the cytokines concentration in the FF, thus making the interpretation of the results and the assessment of the direct impact of endometriosis somewhat challenging.

### 4.2. Summary of Evidence

In our study, we have analyzed 57 factors implicated in the immune and inflammatory response in 87 women undergoing IVF. The univariate analysis, after correction for the risk of type I error, has shown an overexpression of nine cytokines (Figure 2). Out of these nine cytokines, three have already been shown to be increased in the FF (IL1 β [25], IL6 [23], and IL8 [22,24]), two in the peritoneal fluid (MCP-1 [29] and basic FGF (FGF2) [30]), and one in the serum [31] of women with endometriosis. The multivariate analysis (LASSO and logistic regression) allowed us to assess the impact of endometriosis on the aforementioned cytokines while taking into account the potential bio-clinical confounding factors. Our model showed that the two most important variables in women with endometriosis were the downregulation protocol, and MCP-1 (Figure 3), without any dependence between these two variables, thus confirming the importance of MCP-1 in endometriosis.

### 4.3. Biological Rationale

MCP-1 (monocyte chemotactic protein 1), also referred to as CCL2, is produced by different cells, such as endothelial cells, fibroblasts, and immune cells. It plays an important role in the regulation of the migration and infiltration of monocytes, basophiles, lymphocytes T, and natural killer cells in many tissues, including the ovary [32]. Studies have shown a transient increase in the level of MCP-1 in the FF and the ovarian stroma at the time of ovulation [33]. MCP-1 could be implicated in follicular growth, ovulation, and the development as well as the regression of the corpus luteum [34]. These observations have led several researchers to investigate the role of MCP-1 in infertility. For instance, Younis et al. have shown an increase in the serum MCP-1 levels in women with polycystic ovarian syndrome, but found no alteration in women with unexplained infertility or moderate endometriosis before or during ovarian stimulation [35]. Other studies have shown an increase in MCP-1 levels in the peritoneal fluid of women with moderate and severe endometriosis [29,36]. Buyuk et al. found increased serum and FF levels of MCP-1 in obese women undergoing IVF that were negatively correlated with pregnancy rates [37]. The authors postulated that the high levels of MCP-1 could negatively impact the follicular cells’ function and the oocyte quality, by inducing an intracellular proinflammatory state via the activation of MCP-1 receptors or via an influx of monocytes towards the ovary.

Our study included only patients with severe endometriosis, but we identified two subgroups in this patient population: women with FF MCP-1 levels comparable to those in the control group (MCP-1-low group) and women with significantly higher levels (MCP-1-high group) (Figure 4). We found that the cytokine profile was significantly altered in the MCP-1-high group compared to the MCP-1-low and the control group, but we found no difference between the MCP-1-low group and the control group. Moreover, there was a significantly higher basal serum estradiol levels and a significantly lower number of oocytes retrieved in the MCP-1-high group compared to the MCP-1-low and the control groups. Even though some studies have shown an alteration of the estrogen metabolism in the endometriosis lesions [38], serum estradiol levels are usually not modified in women with endometriosis [11,35]. However, in our study, we found an increase in the basal serum estradiol levels in the MCP-1-high subgroup, which could be due to an early follicular recruitment in these patients. Indeed, early recruitment is usually caused by a premature rise of FSH between two menstrual cycles, secondary to a decrease in the ovarian reserve, and is diagnosed by higher serum estradiol and FSH levels early in the cycle [39,40]. Therefore, the MCP-1-high subgroup of patients could have early signs of premature ovarian insufficiency, even though the classic markers of the ovarian reserve (Anti-Müllerian hormone, antral follicle count) are still within the normal range. This observation is further validated by the significantly lower number of oocytes retrieved in these patients despite the higher FSH doses used during controlled ovarian stimulation. In line with the report of Kitajima et al. [41], these findings suggest that there is a more advanced form, or a specific subtype of endometriosis, that is associated with a significant local inflammation causing focal depletion of primordial follicles. The dysregulation of the cytokine profile can lead to a significant alteration of oocyte microenvironment, causing this more advanced or specific from of endometriosis.

### 4.4. Strength and Limitations

The main strength of our study is the homogeneity of our patient population, as we included only patients with severe endometriosis and without any confounding factors. Moreover, we analyzed 57 cytokines and two inflammation-associated factors (MMP-3, Chitinase-3 like 1 protein), more than any previous report in the literature [21,22,23,24,25]. However, we plan to increase our patient population in order to assess whether certain characteristics of the evolution or the severity of the disease (locations, adhesions, previous surgery) are behind the high MCP-1 levels.

## 5. Conclusions

In conclusion, we have found an important biologic heterogeneity in patients with endometriosis, with different levels of MCP-1 expression in the follicular fluid, and we have identified a specific subtype, with a high FF MCP-1 level, that is associated with a significant alteration of the oocyte microenvironment. The existence of several subtypes of endometriosis could explain the conflicting results reported in previous studies. The follicular fluid level of MCP-1 could also be a potential biomarker of endometriosis, and further studies are needed to confirm its role as a potential predictor of the ovarian response to controlled ovarian stimulation in women with endometriosis.

## Figures and Tables

**Figure 1 jcm-09-01306-f001:**
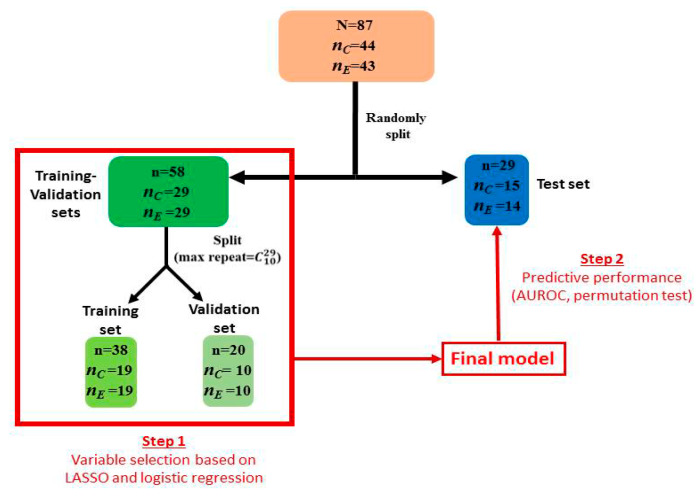
Multivariate analysis was conducted using training-validation (~ 2/3 of all data) and test sets (~1/3 of all data) strategy. Final model (containing variables selected in step 1) predictive capabilities were assessed in the test set to avoid validating overfitted models. Methods used in step 1 included LASSO, then logistic regression. Best models here were selected as those yielding AUROC > 0.95 in the validation set. Variables with no null estimated coefficients were included in a logistic regression model and submitted to a backward procedure. Finally, the best model obtained was applied to the test set to confirm its performance based on the AUROC. Coefficient in the logistic regression were estimated as median of the coefficient of the best models in step 1. Legend: *n_C_*, *n_E_*: number of patients in the control and endometriosis group, respectively; LASSO: Least Absolute Shrinkage and Selection Operator; LR: logistic regression.

**Figure 2 jcm-09-01306-f002:**
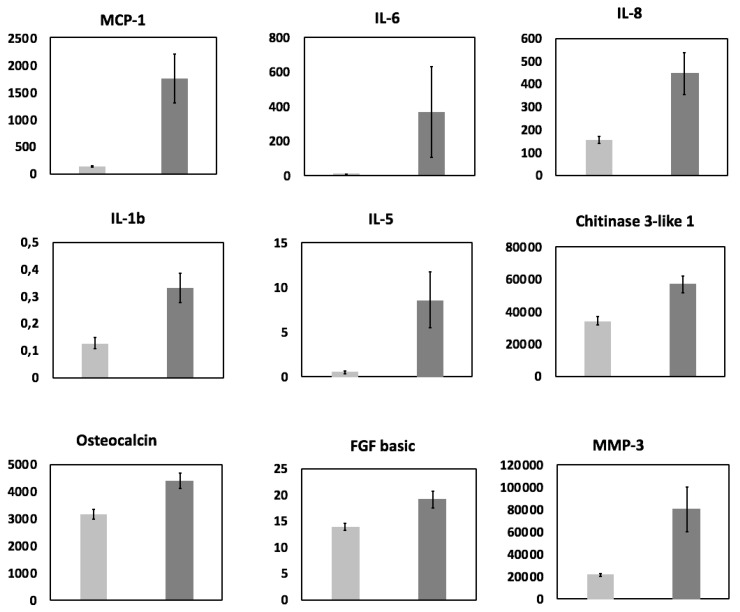
Bar plots representing the follicular fluid concentrations (pg/mL) of the nine cytokines found significantly different between endometriosis (dark gray) and control (light gray) patients. All patients (*n* = 87) were included and data were log-transformed before Student test. Benjamini-Hochberg correction was applied to observed *p*-values for correction of risk I inflation. Error bars represent standard error of the mean (S.E.M.).

**Figure 3 jcm-09-01306-f003:**
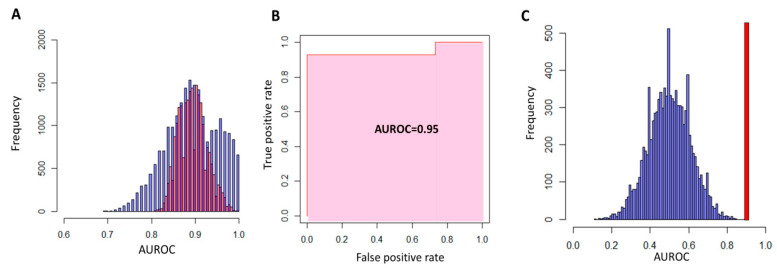
Performances of the LR models in the training-validation sets (**A**) and the test set (**B**,**C**). After LASSO, LR models with downregulation or stimulation protocol and MCP-1 as predictor variables were built and applied to the training and the validation sets. Histograms of AUROC show, as expected, good predictive performance of this models on the training set (red bars in (**A**)). Median AUROC in the validation sets was similar to the one found with the training set (~ 0.90) but exhibited more dispersion (blue bars in (**A**)). When the response vector in the test set was permuted, the performance of the model was indistinguishable from that of the random model (median and mean AUROC equal to 0.5, blue bars in (**C**)) and very different for the predictive capability of the model on the original response vector (AUROC = 0.95, red bar in (**C**)).

**Figure 4 jcm-09-01306-f004:**
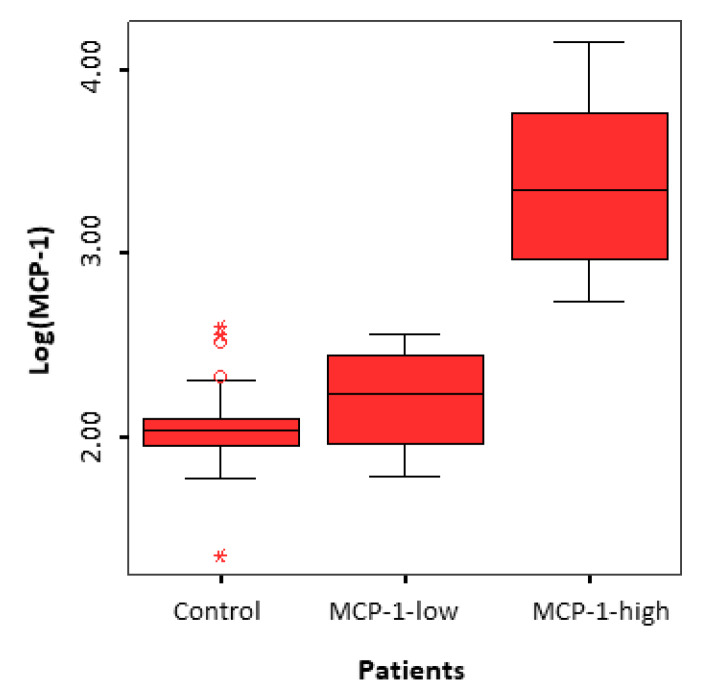
MCP-1 levels (following Log transformation) in the follicular fluid of patients with endometriosis (MCP-1-low and MCP-1-high subgroups) and controls.

**Table 1 jcm-09-01306-t001:** Baseline characteristics of patients and cycles.

Variables		Endometriosis Group (*n* = 43)	Control Group (*n* = 44)	*p*-Value
Age (years)		32.2 ± 3.9	31.4 ± 4.3	0.36
Body Mass Index (kg/m^2^)		23.1 ± 3.6	23.7 ± 4.0	0.46
Tobacco usage Non-smoker Smoker Former smoker Information missing		- 27 3 10 3	- 26 6 11 1	0.62
Baseline FSH (IU/L)		7.40 ± 3.21	8.60 ± 2.68	0.11
Serum Estradiol (pg/mL)		51.5 ± 48.4	36.2 ± 16.1	0.07
Baseline Anti-Mullerian Hormone (AMH) (ng/mL)		2.8 ± 2.08	2.9 ± 1.9	0.78
Antral follicle count		17.6 ± 7.6	17.2 ± 6.8	0.80
Total dose of FSH per cycle (IU)		2810 ± 1072	2375 ± 900	0.04*
Stimulation protocol	Antagonist Agonist	23 20	43 1	< 0.001*
Stimulation	FSH FSH + Luteinizing Hormone (LH)	32 11	35 9	0.57
Treatment type	IVF ICSI	14 29	18 26	0.42
Oocytes retrieved		8.8 ± 7.0	13.4 ± 5.9	0.002*
Embryos per oocytes retrieved (%)		53.5 ± 27.1	52.1 ± 24.5	0.72
Good quality embryos (%)		58.2 ± 31.5	58.5 ± 28.1	0.97

Data are expressed as *n* (%) percentage or mean ± standard deviation. * considered as significant *p*-value < 0.05.

**Table 2 jcm-09-01306-t002:** Characteristics of endometriosis patients, MCP-1-high, and MCP-1-low subgroups.

Variables	Endometriosis Group (*n* = 43)	Subgroup MCP-1-high (*n* = 20)	Sub-group MCP-1-low (*n* = 23)	*p**
Locations of endometriosis Ovaries Fallopian tubes Torus uterinus/uterosacral ligaments Rectovaginal septum Digestive tract Bladder	41 12 25 11 9 5	20 3 12 4 4 1	21 9 13 7 5 4	0.56 0.15 0.82 0.67 - 0.44
Presence of adhesions	14	6	8	0.74
Endometriosis diagnosis made by: Surgery MRI or ultrasound	31 12	15 5	16 7	0.69 0.69
History of surgery: Ovarian Deep endometriosis	19 17	9 8	10 9	0.92 0.95

* *p*-value reflects the comparison between the MCP-1-high and the MCP-1-low groups.

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
