# Peer review of "Elevated Levels of Monocyte Chemotactic Protein-1 in the Follicular Fluid Reveals Different Populations among Women with Severe Endometriosis"

_jcm, 2020, doi:10.3390/jcm9051306_

Round 1
Reviewer 1 Report
This is a prospective observational study looking at whether there are alterations in the cytokine profiles of the follicular fluid (FF) of women with endometriosis undergoing in vitro fertilization. The authors report that MCP-1 (CCL2) was the only variable retained following multivariate analysis. Based on the levels of FF MCP-1 in comparison with a control group without endometriosis, it was possible to separate the endometriosis group into two subgroups, which they identify as MCP-1- (low) and MCP-1+ (high). Those patients in the latter group had altered cytokine profiles in the FF, higher serum estradiol levels and significantly lower numbers of oocytes recovered in comparison with MCP-1- patients or controls. The authors suggest that high FF levels of MCP-1 might identify a subgroup of endometriosis patients with worse prognosis.
This is an interesting and well-designed study that contributes to the understanding of the potential involvement of cytokines/chemokines in endometriosis. The conclusions are warranted by the results and the manuscript appears well written. There are a few relatively minor concerns”:
- Not all of the analytes assayed are cytokines (e.g., MMP-3, Chitinase-3 like 1 protein). It is recommended that the authors refer to these as “cytokines and other inflammation-associated factors”.
- The authors do not use the newer nomenclature for chemokines. At least, they should indicate that MCP-1 is CCL2 the first time it appears.
- The authors indicate on line 158 (Results), that IL-5 levels were uninterpretable, yet they include IL-5 results in Figure 2.
- “FGF basic” should be referred to as “basic FGF (FGF2)”, a more standard nomenclature. “Chitinase-3 like 1” should be referred to as “Chitinase-3 like 1 protein”.
- On lines 204-206, it is indicated that “the concentrations of eleven cytokines (the same ones found different between the two groups, as well as GCSF, sTNFR1, BAFF and osteocalcin) were found significantly higher in the FF of MCP-1+ patients”. Osteocalcin is already included among the nine shown in Figure 2. If we add thre more, that would be twelve and not eleven cytokines. Is this correct?
- Referring to the two subgroups as MCP-1- and MCP-1+ might be somewhat confusing, since those groups have measurable levels of MCP-1 (i.e., are both positive for MCP-1). Perhaps a less confusing definition could be MCP-1-low and MCP-1-high.
- While the rest of the manuscript is well written, the first line in the abstract is somewhat awkward. Perhaps it should read: “To determine if a modification of the cytokine profile occurs in the follicular fluid ….”
Author Response
Dear Editors-in-Chief,
First of all, we would like to thank you for your email. We have carefully read the comments made by the reviewer and the editor. As recommended, we have responded point-by-point to the comments.
Please find below the comments and our point-by-point response.
Reviewer #1:
- “Not all of the analytes assayed are cytokines (e.g., MMP-3, Chitinase-3 like 1 protein). It is recommended that the authors refer to these as “cytokines and other inflammation-associated factors.”
We thank the reviewer for this comment, and we have modified accordingly:
Title of sub-section 2.3, line 108
Line 112-113: “57 cytokines and other inflammation-associated factors (line 112-113)”
- “The authors do not use the newer nomenclature for chemokines. At least, they should indicate that MCP-1 is CCL2 the first time it appears.”
We thank the reviewer for the comment. We mention it on line 262, where we discuss its role in detail: “MCP-1 (monocyte chemotactic protein 1), also referred to as CCL2”. Based on your comment, we have added it the first time it appears, line 176-177: (MCP-1 (also referred to as chemokine ligand 2 (CCL2).
- “The authors indicate on line 158 (Results), that IL-5 levels were uninterpretable, yet they include IL-5 results in Figure 2.”
We thank the reviewer. It is a typing error, the one that’s uninterpretable is IL-15, not IL-5. We have corrected the error on line 160.
“Overall, the GM-CSF and IL-15 levels obtained were uninterpretable (zero),”
- “FGF basic” should be referred to as “basic FGF (FGF2)”, a more standard nomenclature. “Chitinase-3 like 1” should be referred to as “Chitinase-3 like 1 protein.”
We thank the reviewer for the comment and we have modified accordingly, on lines 114, 164-165 and line 255.
- On lines 204-206, it is indicated that “the concentrations of eleven cytokines (the same ones found different between the two groups, as well as GCSF, sTNFR1, BAFF and osteocalcin) were found significantly higher in the FF of MCP-1+ patients”. Osteocalcin is already included among the nine shown in Figure 2. If we add three more, that would be twelve and not eleven cytokines. Is this correct?
We thank the reviewer for pointing out this error. In fact, it is a typing error. There are only 11 cytokines: the 9 previously mentioned, but without osteocalcin (which was not different between the two groups), with GCSF, sTNF_R1 and BAFF. We have modified the sentences accordingly: “After Benjamini-Hochberg correction, the concentrations of eleven cytokines (the same ones found different between the two groups barring osteocalcin, as well as GCSF, sTNF_R1 and BAFF)”.
- Referring to the two subgroups as MCP-1- and MCP-1+ might be somewhat confusing, since those groups have measurable levels of MCP-1 (i.e., are both positive for MCP-1). Perhaps a less confusing definition could be MCP-1-low and MCP-1-high.
We thank the reviewer for the comment and we have modified accordingly in the entirety of the manuscript
- While the rest of the manuscript is well written, the first line in the abstract is somewhat awkward. Perhaps it should read: “To determine if a modification of the cytokine profile occurs in the follicular fluid ….”
We thank the reviewer and we have changed the sentence as suggested.

Reviewer 2 Report
The article by Bouet et al focuses on the investigation of the role of the elevated levels of monocyte chemotactic protein-1 in the follicular fluid (FF) of women with endometriosis undergoing in vitro fertilization (IVF). Thus, a group of 87 women undergoing IVF was subdivided in two groups; the first one included 43 cases with severe endometriosis-related infertility and the second one consisted of 40 controls with other causes of infertility. The inflammatory profile of the FF was analyzed, considering that it is a direct reflection of the oocyte micro-environment, playing a crucial role in the acquisition of gamete competence. The authors identified two subgroups in the patient population, women with FF MCP-1 levels that were comparable to those in the control group and women with significantly higher levels. The second subgroup (with high MCP-1 levels) was found to be associated with a significant alteration of the oocyte microenvironment, thus representing a potential predictor of the ovarian response to the ovarian stimulation in women with endometriosis.
The language used is concise and easy to understand, enabling even non-native English speakers or young researchers to receive critical information. The scientific questions posed are very clear, methodology used and the statistics are sufficient, while the figures and tables confer to a better understanding of the article.
Overall, this text is a very interesting article, which may be proven helpful to both clinicians and researchers, and thus is highly publishable.
Author Response
Dear Editors-in-Chief,
First of all, we would like to thank you for your email. We have carefully read the comments made by the reviewer and the editor. As recommended, we have responded point-by-point to the comments.
Please find below the comments and our point-by-point response.
Reviewer #2:
- “The language used is concise and easy to understand, enabling even non-native English speakers or young researchers to receive critical information. The scientific questions posed are very clear, methodology used and the statistics are sufficient, while the figures and tables confer to a better understanding of the article. Overall, this text is a very interesting article, which may be proven helpful to both clinicians and researchers, and thus is highly publishable.”
We thank the reviewer for his critical appraisal.
